# Preparation and Characterization of Model Tire–Road Wear Particles

**DOI:** 10.3390/polym14081512

**Published:** 2022-04-08

**Authors:** Chae Eun Son, Sung-Seen Choi

**Affiliations:** Department of Chemistry, Sejong University, 209 Neungdong-ro, Gwangjin-gu, Seoul 05006, Korea; chaeunmcl@sju.ac.kr

**Keywords:** tire–road wear particle (TRWP), tire wear particle (TWP), mineral particle, model TRWP, SEM, TGA

## Abstract

Tire tread wear particles (TWPs) are one of major sources of microplastics in the environment. Tire–road wear particles (TRWPs) are mainly composed of TWPs and mineral particles (MPs), and many have long shapes. In the present work, a preparation method of model TRWPs similar to those found in the environment was developed. The model TRWPs were made of TWPs of 212–500 μm and MPs of 20–38 μm. Model TWPs were prepared using a model tire tread compound and indoor abrasion tester while model MPs were prepared by crushing granite rock. The TWPs and MPs were mixed and compressed using a stainless steel roller. The TWPs were treated with chloroform to make them stickier. Many MPs in the model TRWP were deeply stuck into the TWPs. The proper weight ratio of MP and TWP was MP:TWP = 10:1, and the double step pressing procedure was good for the preparation of model TRWPs. The model TRWPs were characterized using scanning electron microscopy (SEM) and thermogravimetric analysis (TGA). The model TRWPs had long shapes and the MP content was about 10%. The model TRWPs made of TWPs and asphalt pavement wear particles showed plate-type particles deeply stuck into the TWP. Characteristics of model TRWPs can be controlled by employing various kinds and sizes of TWPs and MPs. The well-defined model TRWPs can be used as the reference TRWPs for tracing the pollutants.

## 1. Introduction

Tire tread wear is one of principal non-exhaust dust sources produced from road traffic along with brake and pavement wear [1,2,3,4,5,6,7,8,9,10]. Abraded tire tread material without other particles is called tire tread wear particles or tire wear particles (TWP), and TWP combined with other particles including road wear materials deposited on the road is called tire–road wear particle (TRWP) [2,4,8,9,11]. Tire tread contacts with the road surface including lots of mineral particles (MPs) during driving and it is abraded by friction with the road surface. During the wear process, the tire tread interacts with many MPs and some MPs will be stuck into the rubber to produce TRWPs.

TRWPs are deposited on the road and transported to soils near the roadside or to surface waters, and they are widely found in the environment [2]. Traffic-related non-exhaust emissions including TRWPs are also considered as a principal contribution to particulate matter (PM) in ambient air [6,7,12]. As to aquatic environments, TRWPs have been addressed in connection with microplastic pollution [13,14,15,16,17,18]. The TRWPs have higher density than seawater and they can settle as sediments in the sea and in rivers [19].

Polymers are very important organic compounds to change human lifestyle, but microplastics are a contaminant of increasing ecotoxicological concern in environments [15,20]. Tire tread compounds are usually made of rubber, filler (carbon black and silica), curatives, and other additives. TWPs are generated by friction of the tire tread and road surface. Abrasion rate of tire tread on a cement road surface was larger than that on an asphalt one [21]. TWP contents in road dust were less than 5 wt% [22]. TRWP can be separated from the road dust using density separation [4,23].

TRWP has been considered as one of the major sources of microplastics [2,3,4]. Well-defined TRWPs are required for tracing a study on the traveling path of TRWPs from road to river and sea to accumulate as sediment. Investigation on the changing process to micro dust from TRWPs by aging and breaking are also required for environmental effects evaluation of TRWPs. Well-defined TRWPs are also needed for study on the environmental effect evaluation and the risk for human health. Real TRWPs have a variety of rubber compositions, different kinds and contents of fillers, and a great variety of kinds and contents of inorganic and organic components. The MP content in the TRWP is also very varied [1]. Thus, well-defined TRWPs are essential for reliable research on TRWPs in many different fields. The objective of this study is the development of preparation methods of reliable and homogeneous model TRWPs with similar shapes to real ones. TWPs were prepared by creating wear on a model tread vulcanizate and MPs were prepared by breaking granitic rock. The model TRWPs were made of the TWPs and MPs by mixing and compressing with a stainless steel roller. Target sizes of the model TRWPs were 212–500 μm because sizes of real TRWPs produced on road are below 500 μm [24]. The model TRWPs were characterized using scanning electron microscopy (SEM) and thermogravimetric analysis (TGA).

## 2. Materials and Methods

### 2.1. Materials

Carbon black-filled natural rubber (NR) vulcanizate (Appendix A) was used for the preparation of TWPs. The NR vulcanizate was abraded using laboratory abrasion tester of LAT 100 (VMI group, Epe, the Netherlands). The wear particles were separated by size using a sieve shaker Octagon 200 (Endecotts Co., London, UK). TWPs of 212–500 μm were employed for the preparation of model TRWPs. MPs were prepared by crushing a granitic rock (Pocheonseok supplied from Dongastone Co., Pocheon, Korea) into powder with a hammer and by following separation by size using the sieve shaker. MPs of 20–38 μm was used for the preparation of model TRWPs. Chloroform was purchased from Aldrich Co. (St. Louis, MO, USA). Figure 1 shows SEM images of the TWPs and MPs.

### 2.2. Sampling of Real TRWP

Real TRWPs were collected from road dust at a bus stop. Road dust accumulated between the curb and road was gathered by sweeping with a broom, and was separated by size using a sieve shaker Octagon 200 (Endecotts Co., London, UK). TRWPs of 212–500 μm were picked out from the road dust. Shapes of the TRWPs were observed using a scanning electron microscope (CUBE-II tabletop SEM, Emcrafts Co., Gwangju-si, Korea).

### 2.3. Preparation of Model TRWP

Model TRWPs were prepared as follows (Appendix A): (1) Put the TWPs of 212–500 μm into a stainless steel container. (2) Add the MPs of 20–38 μm on the TWPs. (3) Mix the TWPs and MPs with a spatula. (4) Press the mixture with a stainless steel roller. (5) Separate the TRWPs of 212–500 μm from the mixture using the sieve shaker. Mixing ratio (weight) of TWP and MP was 1:2, 1:5, and 1:10. Two types of the TWPs were employed: (1) TWPs without any treatment and (2) TWPs treated with chloroform. The chloroform-treated TWPs were prepared by soaking the untreated TWPs in chloroform at room temperature for 1 h and by drying at room temperature for another 1 h.

Single and double step pressing procedures were employed. The single step pressing procedure was pressing the mixture of TWPs and MPs with a stainless roller 200 times. The double step one was that the first obtained TRWPs of 212–500 μm were separated and then the second rolling of 200 cycles was carried out again.

### 2.4. Characterization of the Model TRWP

The model TRWPs of 212–500 μm were selected by sieving and their characteristics were analyzed. Shapes of the model TRWPs were observed using a scanning electron microscope (CUBE-II tabletop SEM, Emcrafts Co., Gwangju-si, Korea). The model TRWPs were analyzed by a thermogravimetric analyzer to determine contents of the TWP and MP. TGA was carried out using a thermogravimetric analyzer (TGA/DSC1 star system of Mettler Toledo Co., Giessen, Germany). The sample was heated from 50 to 900 °C at the heating rate of 20 °C/min. TGA was performed under inert (N_2_) and oxidative (air) conditions at 50–700 °C and 700–900 °C ranges, respectively. The gas flow was 50 mL/min. Elemental analysis was performed using energy-dispersive spectroscopy (EDS) (Hitachi SU-8010, Hitachi Co., Hitachi Co., Tokyo, Japan). The accelerating voltage was 15 kV.

## 3. Results and Discussion

### 3.1. Characteristics of Real TRWPs and MPs

Shapes of real TRWPs collected on the road were analyzed using SEM. Many TRWPs had long shapes (aspect ratios of 2–6) similar to sausages and slightly rough surfaces (Figure 2). There are lots of various MPs on the TWP surface. Some MPs are deeply stuck into the rubber and others are adsorbed on the surface. Sizes of the MPs on the TRWPs are very various from below 1.0 μm to above 10.0 μm (the big image in Figure 2). Based on the long shapes of real TRWPs, we tried to make model TRWPs with long shapes. There are lots of MPs in the road particles (Appendix A). The MPs on the road can be crushed into smaller ones again by friction with tires of buses and passenger cars and then they will be stuck into or be adsorbed on the TWPs. Hence, TRWPs can be made by pressing the TWPs and MPs with a roller. By treating the real TRWPs with chloroform, their surfaces are rougher and more porous (Appendix A). Thus, if the model TWPs are treated with chloroform their surfaces will be rougher and have more cavities.

Elemental analysis of the real MPs (rMP1, rMP2, and rMP3) was performed using energy-dispersive spectroscopy (EDS). Besides metal elements and oxygen (O), carbon (C) was detected (Appendix A). The carbon would come from exhaust gas and bitumen of asphalt pavement. Major elements of the real MPs were Na, Al, Si, K, and Ca. Element components of the rMP1, rMP2, and rMP3 were different from each other. Elemental analysis of the real TRWPs was also carried out after chloroform treatment to remove some MPs on the surface (Appendix A). Carbon and oxygen should originated from rubber and metal oxide, respectively. Besides the various metal elements, there were specific elements of sulfur (S) and zinc (Zn) originating from elemental sulfur (S_8_) and zinc oxide (ZnO), respectively, used as curatives of rubber vulcanizates [25,26]. Exitence of S and Zn is a concrete evidence of TWP.

### 3.2. Influence of Mixing Ratio of the MP and TWP on Property of TRWP

Model TRWPs were made by varying the mixing ratio (weight) of the MP and TWP to examine the proper mixing ratio. Size distributions of the model TRWPs made by the mixing ratios of 2/1, 5/1, and 10/1 (MP/TWP) were summarized in Table 1. Untreated TWPs were used. For the mixing ratio of 2/1, the TRWPs above 500 μm were about 60% and those of 212–500 μm were only 2.9%. Considering the initial content of TWP of 33.3 wt%, content of the TRWP of 212–500 μm (2.9%) is very low. The TRWPs above 500 μm might be generated by clustering the TWPs. For the mixing ratio of 5/1, the TRWPs of above 500 μm were only 2.3% and those of above 1000 μm were not observed. Content of the TRWPs of 212–500 μm was 21.5%, which means that most TWPs participated in production of TRWPs considering the initial content of TWP of 16.7 wt%. Particles of below 38 μm were 74.8% and they must be the MPs. Particles of below 20 μm should be MPs crushed again by pressing. For the mixing ratio of 10/1, the TRWPs of above 500 μm were only 1.5% and those of 212–500 μm were 10.7%, which implies that most TWPs participated in production of TRWPs considering the initial content of TWP of 9.1 wt%. Particles of below 38 μm were 87.1%. Thus, this can lead to a conclusion that the proper initial MP/TWP ratio is 10/1.

### 3.3. Influence of the Pressing Procedures on Properties of Model TRWP

In order to make the shape of mineral incrustations on the TWP and to embed the MPs into the TWP, mixture of the TWPs and MPs was pressed with a stainless steel roller. Single and double step pressing procedures were employed to examine the influence of the pressing procedures on properties of the TRWPs. The single step pressing procedure is composed of one rolling 200 times. There were lots of the MPs on the TWP in the SEM images of the TRWPs made by the single step one (Appendix A), but most MPs were weakly bound to the TWP and the TRWPs had a rough surface. Shapes of the TRWPs were not even and their aspect ratios became smaller compared to the initial TWPs. Sizes of the MPs became much smaller by pressing and most MPs were below 10 μm sizes (the big image in Appendix A). The TRWPs made using the untreated TWPs and by the single step pressing procedure were different from the real ones. This might be because of weak interaction between the untreated TWPs and the MPs.

The model TRWPs made of the untreated TWPs by the double step pressing procedure had long shapes similar to the real ones though the aspect ratios were relatively smaller (Appendix A). Aspect ratios of the TRWPs were 2.5–5.0. Some MPs were relatively deeply stuck into the TWP. The MPs were more crushed into smaller pieces and most MPs were below 10 μm. SEM image of the TRWP surface showed uneven surface and weak interactions between the TWPs and MPs. Cavities in the TWP were notably decreased but the TRWPs were still different from the real ones.

Component ratios of the TWP and MP in the TRWPs were analyzed using TGA. TGA thermogram of the TWP shows the ash content of 4.6% which is coming from zinc oxide and mineral components in the NR (Appendix A). Ash contents of the TRWPs made of untreated TWPs by the single and double step pressing procedures were much greater than that of the TWP (Appendix A). The extra ash contents must be the MPs. Ash content of the TRWP prepared by the double step pressing procedure was greater than that of the TRWP prepared by the single one. In the SEM images of Appendix A, more MPs were shown in the TRWP prepared by the single step pressing procedure than that prepared by the double one, but the TGA results were inverse. This implies that MPs deeply stuck into the TWP in the TRWP prepared by the double step pressing procedure were more than the TRWP prepared by the single one. The detail TGA results were summarized in Table 2.

The ash contents of the TRWPs prepared by the single and double step pressing procedures were 16.8 ± 1.2 and 17.2 ± 1.3%, respectively. By correcting the ash content of the TWP, the MP contents were 12.2 and 12.6%, respectively. The MP contents of the model TRWPs were lower than those of the real ones. The MP contents of the real TRWPs were above 20% [1]. However, the MP contents of the real TRWPs were very variable of 23–34%. The MP content ranges of the model TRWPs were relatively very narrow as listed in Table 2. Thus, we believe that it is possible to make reliable and homogeneous TRWPs using the preparation method developed in this work.

### 3.4. Chloroform Treatment Effect of TWP on Properties of TRWP

In order to make the TWPs sticky, the TWPs were treated with chloroform. The chloroform-treated TWPs had relatively wet and soft surfaces, while the untreated ones had dry and rough surfaces (Appendix A). The TRWPs made of the chloroform-treated TWPs by the single step pressing procedure also had rough surfaces with some cavities and most MPs were weakly bound to the TWP surface (Appendix A). Aspect ratios of the TRWPs as well as the MP sizes decreased by pressing with a roller. Compared to the TRWPs made of the untreated TWPs, some MPs were more deeply stuck into the TWP and there were various sizes of MPs. 

Figure 3 shows SEM images of the TRWPs made of the chloroform-treated TWPs and by the double step pressing procedure. The second rolling step was applied to only the first made TRWPs of 212–500 μm. The aspect ratios were largely decreased but the shapes looked like the real TRWPs. Many MPs were deeply stuck into the TWP, most cavities in the TWP disappeared, and there were MPs of various sizes on the TRWP but MPs of above 10 μm were few (the big image in Figure 3). The TRWPs made of the chloroform-treated TWPs and by the double step pressing procedure were more similar to the real ones than the others. This might be due to the increased sticky property of TWPs by the chloroform treatment. The increased sticky property can lead to enhancing interactions between the MPs and TWP, and the MPs may be more deeply stuck into the rubber. In addition to this, the second rolling step can lead to better interactions between the TWP and MPs and removing weakly bound MPs from the TWP. These processes can also reflect wear mechanism of tire tread on the road.

Figure 4 shows the TGA thermograms of the TRWPs made of the chloroform-treated TWPs by the single and double step pressing procedures. Difference in the component ratios of the TRWP prepared by the single and double step pressing procedures was not large. Ash content of the TRWP prepared by the single step pressing procedure was relatively greater than that of the TRWP prepared by the double one. This is because the second rolling process is performed only with the first made TRWPs. The TGA results were summarized in Table 3. Tolerance ranges for the TRWPs made by the double step pressing procedure were, on the whole, lower than those made by the single one. Especially, the ash content of the TRWPs made by the double step pressing procedure showed very low tolerance. This means that preparation of model TRWPs using the chloroform-treated TWPs and the double step pressing procedure can give more uniform TRWPs. Lower ash content for the TRWPs made by the double step pressing procedure might be due tot some loosely bound MPs being detached from the TRWP surface through the second rolling step. 

### 3.5. Preparation of Model TRWPs Using Asphalt Pavement Wear Particles

During driving on asphalt pavement road, both tire tread and asphalt pavement are abraded by friction between them. Other model TRWPs were made using asphalt pavement wear particles (20–38 μm) and the TWPs. The double step pressing procedure was applied. Asphalt pavement wear particles were prepared using a model asphalt pavement (Appendix A). The asphalt pavement wear particles (APP1 and APP2) had high contents of carbon and oxygen which come from bitumen and metal oxide components (Appendix A). The major mineral components are Al, Si, and Ca, but mineral components will be different depending on the kinds of stones used for the asphalt pavement. Elemental compositions of the asphalt pavement wear particles are different from each other.

Shapes and surfaces of the TRWPs made of the untreated TWPs and the asphalt pavement wear particles looked like the TRWPs made of the MPs, and some large-sized particles were observed on the surface (Appendix A). However, the TRWPs made of the chloroform-treated TWPs and the asphalt pavement wear particles had different surfaces, and there were some plate-type particles which were marked by the yellow circle and arrow in Figure 5. The plate-type particles are also observed in real TRPWs. Hence, plate-type particles in real TRWPs might be asphalt pavement wear particles.

### 3.6. Proper Preparation Process of Model TRWPs Similar to Real TRWPs

A proper preparation procedure of well-defined model TRWPs is as follows (Figure 6): (1) Model TWPs are prepared by abrading a model rubber specimen using a laboratory abrasion tester and by selecting the proper sizes using a sieve shaker, and the selected TWPs were treated with chloroform. (2) Model MPs are prepared by breaking a stone with a hammer and by selecting the proper sizes using a sieve shaker. (3) The TWPs and MPs are mixed in a 1:10 ratio and pressed with a stainless steel roller. (4) The first made TRWPs are obtained by removing the smaller and larger particles than the initial TWPs. (5) Finally, the first made TRWPs are pressed again with a stainless steel roller and the final model TRWPs are obtained by removing the smaller particles then the initial TWPs. The model TRWPs are characterized by their morphlogy and the component content ratios using SEM and TGA, respectively.

Properties of TRWPs depend on their sizes, the rubber composition, and the kinds and contents of the inorganic and organic components. Well-defined TRWPs are needed for studying the evaluation of environmental effects and the risk for human health. It is possible to make well-defined TWRPs when using TWPs and MPs (Appendix A). Various target TWP materials can be prepared by controlling the rubber compound formulation and cure system, while various stones, glasses, and pavements may be also used as target MPs. The target TWPs are prepared by abrasion and a following separation, while the target MPs are prepared by crushing or abrasion and a following separation. The target TWPs can be treated with chloroform as occasion demands. 

### 3.7. Variety of Applicable TWPs and MPs

There are many TWPs and MPs to apply for the preparation of model TRWPs and the manufacturing methods are also various. Besides laboratory abrasion testers, a cryogenic crusher (cryogenic grinder) and a hand drill can be used for preparation of model TWPs. Major merits for application of a cryogenic crusher and a hand drill are ease of use and no limitation in kinds and shapes of target rubber samples. Using a cryogenic crusher or a hand drill is very simple, and prices of the instruments are not expensive. A cryogenic crusher and a hand drill do not require any specific specimen, whereas abrasion testers usually need their own specimens. Appendix A shows SEM images of various TWPs with different rubber compositions and carbon black contents made by cryogenic crushing. When using a hand drill, rubber wear particles for model TWPs can be prepared just by installing an abrasive tip in the hand drill and operating the drill on the target rubber sample (Appendix A).

Model MPs can be easily prepared by breaking a target sample such as a model concrete pavement and a specific stone (Appendix A). For preparation of concrete pavement wear particles, a model concrete pavement can be used. The concrete pavement wear particles (CPP1 and CPP2) have high contents of oxygen and calcium which come from metal oxide components and calcium carbonate. They have lower carbon contents than the asphalt ones. Besides Ca, major mineral components are Mg, Al, Si, and Fe. Sulfur can be detected as a special element in the concrete pavement wear particles. Compositions of stone wear particles (SP1 and SP2) are different according to the parts. The stone wear particles have lower carbon contents than the asphalt and concrete pavement wear particles.

## 4. Conclusions

Real TRWPs have various shapes but long shapes are common. There are lots of MPs on the TWP surface in real TRWPs and some MPs are deeply stuck into the TWPs. Contents and sizes of MPs in real TRWPs are very varied. A method to prepare model TRWPs similar to those found in the environment was developed using the TWPs of 212–500 μm and the MPs of 20–38 μm. The proper mixing ratio was found to be 10/1 of MP/TWP. The double step pressing procedure was better than the single one for the preparation of model TRWPs. The second rolling step made interactions between the TWP and MPs stronger and weakly bound MPs were removed from the TWP by the second rolling step. The MP content ranges of the model TRWPs were relatively very narrow compared to real ones. By employing chloroform-treated TWPs, more MPs were deeply stuck into the TWPs due to the increased sticky property of TWPs. The ash content for the TRWPs made by the double step pressing procedure showed very low tolerance. The MP contents of the model TRWPs made of the chloroform-treated TWPs and by the double step pressing procedure were above 10% and the tolerance was narrow. Model TRWPs made of the asphalt pavement particles showed some plate-type particles on the surface similar to real ones. Kinds of applicable TWPs and MPs are varied and the manufacturing methods are also varied. Besides laboratory abrasion testers, a cryogenic crusher (cryogenic grinder) and a hand drill could be used for preparation of TWPs. Preparation and characterization procedure of model TRWPs was as follows: (1) Size-selected chloroform-treated TWPs (212–500 μm) and MPs (20–38 μm) are mixed with the mixing ratio of 1/10. (2) The mixture is first pressed with a stainless steel roller. (3) TRWPs of 212–500 μm are selected and the second rolling step is performed. (4) Finally, TRWPs of 212–500 μm are selected and SEM and TGA analyses are performed for characterization of the TRWPs. The well-defined model TRWPs could be obtained using the preparation method developed in this study. Model TRWPs can be used for tracing the study of TRWPs produced on the road to sediment of rivers and seas and the risk to human health. The ways to reduce generation of TWPs would include the development of high abrasion-resistant tread compounds and improvement of road conditions.

## Figures and Tables

**Figure 1 polymers-14-01512-f001:**
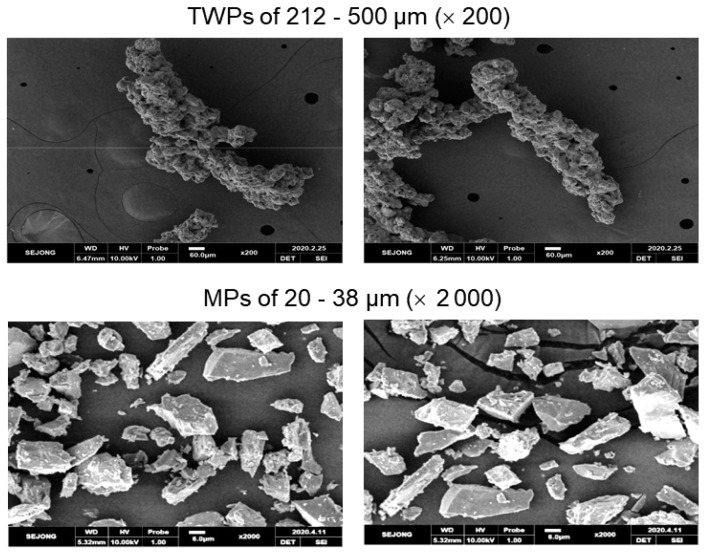
SEM images of the model TWPs of 212–500 μm and the model MPs of 20–38 μm. The scale bars are 60 and 6 μm for the magnifications of 200 and 2000, respectively.

**Figure 2 polymers-14-01512-f002:**
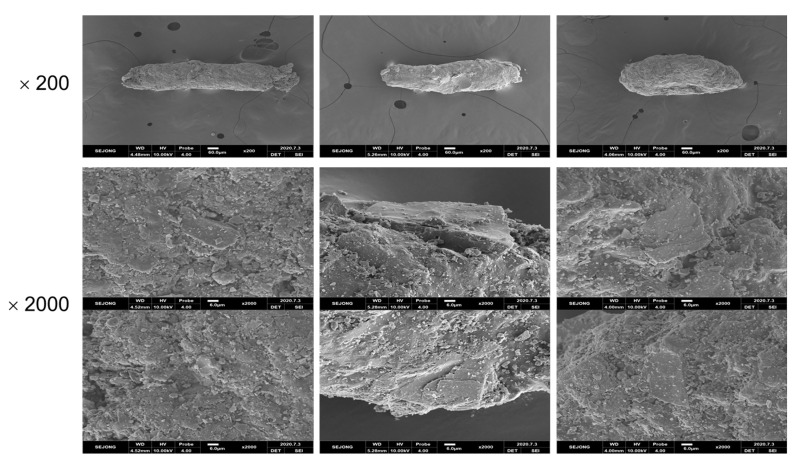
SEM images of the real TRWPs of 212–500 μm collected at a bus stop. The scale bars are 60 and 6 μm for the magnifications of 200 and 2000, respectively.

**Figure 3 polymers-14-01512-f003:**
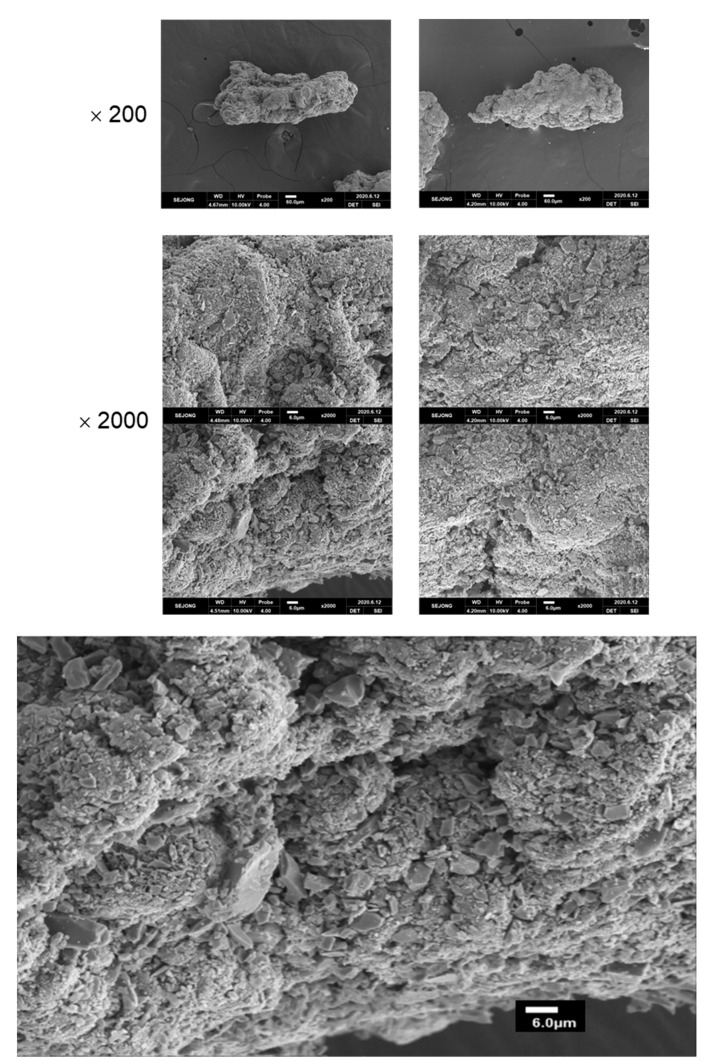
SEM images of the model TRWPs made of the chloroform-treated TWP by the double step pressing procedure. The scale bars are 60 and 6 μm for the magnifications of 200 and 2000, respectively.

**Figure 4 polymers-14-01512-f004:**
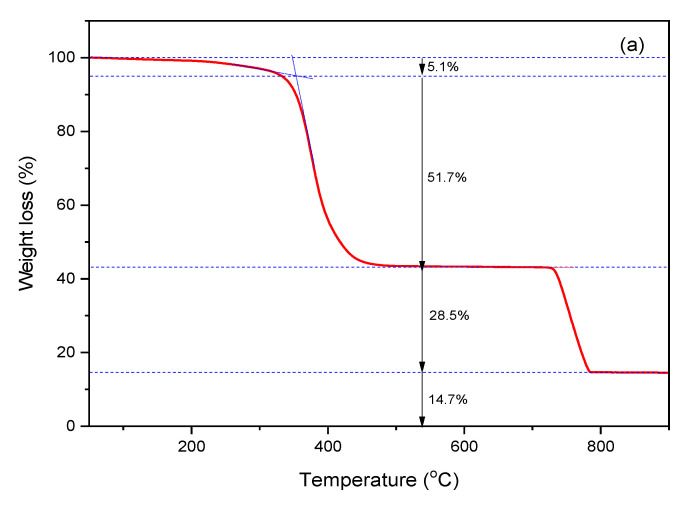
TGA thermograms of the model TRWPs made of the chloroform-treated TWP by the single (**a**) and double (**b**) step pressing procedures.

**Figure 5 polymers-14-01512-f005:**
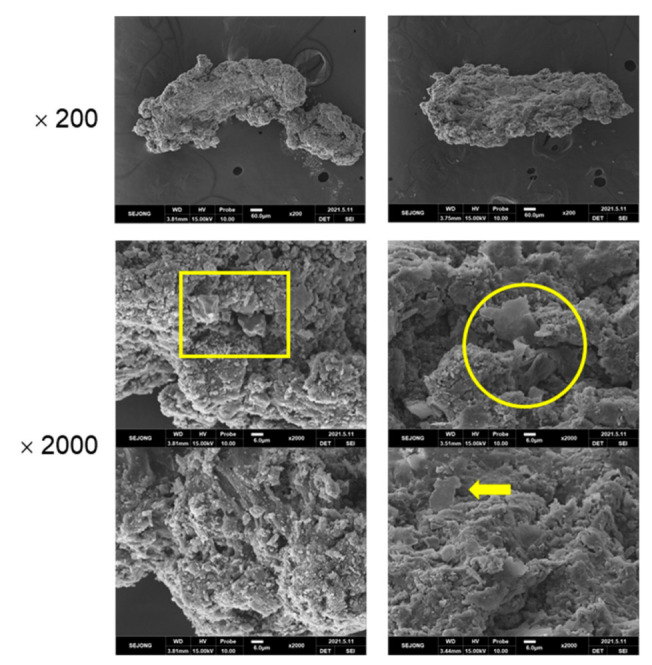
SEM images of the model TRWPs made of the chloroform-treated TWPs (212–500 μm) and asphalt pavement wear particles (20–38 μm). The scale bars are 60 and 6 μm for the magnifications of 200 and 2000, respectively.

**Figure 6 polymers-14-01512-f006:**
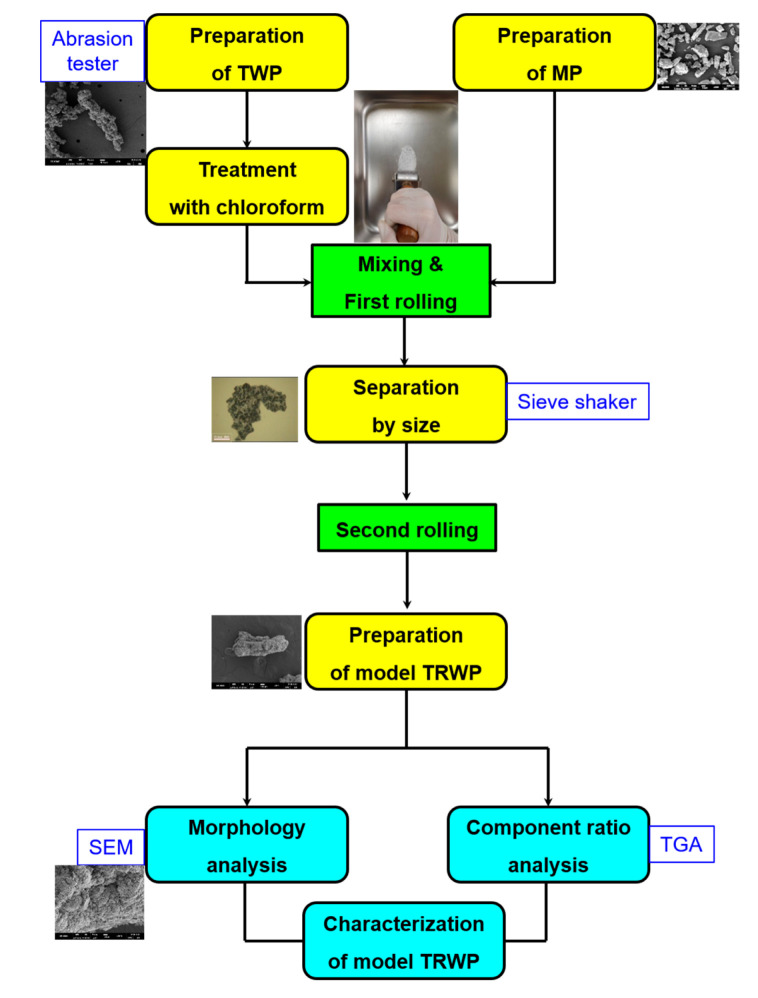
Preparation and characterization process of model TRWPs.

**Table 1 polymers-14-01512-t001:** Particle size distribution of mixture of initial particles (MP and TWP) and TRWPs made by different mixing ratios of the MP and TWP (%).

Sieve Size (μm)	Mixing Ratio of MP/TWP (Weight)		
	2/1	5/1	10/1
1000	7.4	-	-
500	52.3	2.3	1.5
212	2.9	21.5	10.7
106	2.5	-	0.5
63	7.6	-	0.2
38	9.2	1.4	-
20	18.1	63.8	74.8
<20	-	11	12.3

**Table 2 polymers-14-01512-t002:** TGA results of the model TRWPs prepared by the TWPs without treatment.

Sample No.	Component Ratio (%)			
	Volatile Component	Polymer	Carbon Black	Ash
Single step pressing				
1	6.8	51.7	25.5	16.0
2	6.0	52.5	25.2	16.3
3	8.0	49.6	24.2	18.2
Average	6.9 ± 1.0	51.3 ± 1.5	25.0 ± 0.7	16.8 ± 1.2
Double step pressing				
1	7.1	51.7	25.4	15.8
2	6.9	50.4	25.1	17.6
3	4.9	51.3	25.5	18.3
Average	6.3 ± 1.2	51.2 ± 0.7	25.3 ± 0.2	17.2 ± 1.3

**Table 3 polymers-14-01512-t003:** TGA results of the model TRWPs made of the chloroform-treated TWPs.

Sample No.	Component Ratio (%)			
	Volatile Component	Polymer	Carbon Black	Ash
Single step pressing				
1	3.8	49.7	25.6	20.9
2	5.1	51.7	28.5	14.7
3	5.0	53.0	28.8	13.2
Average	4.3 ± 0.8	51.5 ± 1.7	27.6 ± 1.8	16.3 ± 4.1
Double step pressing				
1	3.9	57.4	25.1	13.6
2	5.5	52.5	28.4	13.6
3	4.9	52.9	28.2	14.0
Average	4.8 ± 0.8	54.3 ± 2.7	27.2 ± 1.9	13.7 ± 0.2

## Data Availability

The data presented in this study are available on request from the corresponding author.

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
