# Peer review of "Preparation and Characterization of Model Tire–Road Wear Particles"

_polymers, 2022, doi:10.3390/polym14081512_

Round 1

Reviewer 1 Report

This manuscript deals with the preparation and characterization of model tire-road wear microparticles, aiming at reduce the environmental impact of this hybrid pollutant. The topic is very interesting but the text suffers from some problems that deserve care. The Introduction is too short and too focalized on the specific problem, not considering the framework in which this problem is inserted, such as that of pollutant persistence on the environment (I suggested adding some more lines about this statement by using the suggestions reported in the attached .pdf). The preparation description must be for a scientific journal and not for a laboratory report (see attached report). The experimental deserves revision (see attached report), as well as some aspects related to the decimal place, in my opinion it is meaningless to report decimal place if the standard deviation is higher than 1. The manuscript needs also a revision to eliminate many typos.

Author Response

Q1. Line 17: Define abbreviation first time is used.

A1. Full names of SEM and TGA were added.

Q2. Lines 41-43: In my opinion the Introduction it is too short and should be implemented by writing a few lines about the persistence of plastic in general in the environment. The authors can be inspired by these researches, like for instance, [Blanco, I. Lifetime Prediction of Polymers: To Bet, or Not to Bet—Is This the Question? Materials 2018, 11, 1383. https://doi.org/10.3390/ma11081383 Zeynep Akdogan, Basak Guven. Microplastics in the environment: A critical review of current understanding and identification of future research needs. Environmental Pollution Volume 254, Part A, 2019, 113011]. Before this sentence is the right place to discuss about this.

A2. The sentences “Polymers are very important organic compounds to change human lifestyle, but microplastics are a contaminant of increasing ecotoxicological concern in environments [20,21]. TRWP has been considered as one of major sources of microplastics [2-4].” were added before Line 40. And two new references of 20 and 21 were also added.

  1. Blanco, I. Lifetime Prediction of Polymers: To Bet, or Not to Bet - Is This the Question?: Materials 2018, 11, 1383.
  2. Akdogan, Z.; Guven, B. Microplastics in the environment: A critical review of current understanding and identification of future research needs: Environ. Pollut. 2019, 254, 113011.

Q3. Line 63: maybe is crusching.

A3. “cruching” was changed to “crushing”.

Q4. Line 71: avoid repetition.

A4. “at a bus stop” was deleted.

Q5. Lines 73-75: move this part in the characterization.

A5. That part is about characterization of real TRWPs. ‘2.4. Characterization of the Model TRWP’ is about the model TRWP not real one.

Q6. Lines 78-81: Considering this is not a report I would suggest to the Authors to explain not in this way (third person and list), but in a discursive way, as it is proper to a scientific paper.

A6. That is about preparation procedure of model TRWP. We think that detail preparation procedure of model TRWP is needed for the sake of readers.

Q7. Line 86: improve English.

A7. The sentence “The single step pressing procedure was that the mixture of TWPs and MPs was pressed with a stainless roller 200 times.” was changed to “The single step pressing procedure was pressing the mixture of TWPs and MPs with a stainless roller 200 times.”.

Q8. Line 93: (TGA) is not the acronym for thermogravimetric analyzer but for thermogravimetric analysis.

A8. “(TGA)” was deleted.

Q9. Line 102: please explain better what are 2-6?

A9. The aspect ratio is the ratio of the major and minor axes. The aspect ratios of 2-6 mean that the major axis is longer 2-6 times than the minor one so the TRWPs have long shapes.

Q10. Line 117 and 122: do you mean Elemental analysis?

A10. “Element analysis” was changed to “Elemental analysis”.

Q11. Line 118: why this technique was not described in the experimental section?

A11. The EDS analysis conditions “Elemental analysis was performed using energy-dispersive spectroscopy (EDS) (Hitachi SU-8010, Hitachi Co., Japan). The accelerating voltage was 15 kV.” was added in the experimental section.

Q12. Lines 126-127: please cite reference

A12. Two new references of 26 and 27 were added.

  1. Ryu, G.; Kim, D.; Song, S.; Lee, H. H.; Ha, J. U.; Kim, W. Wear particulate matters and physical properties of silica filled ENR/BR tread compounds according to the BR contents: Elast. Compos. 2021, 56, 243-249.
  2. Lee, G.-B.; Shin, B.; Han, E.; Kang, D.; An, D. J.; Nah, C. Effect of blade materials on wear behaviors of styrene-butadiene rubber and butadiene rubber: Elast. Compos. 2021, 56, 172-178.

Q13. Tables 2 and 3: what is the meaning of reporting decimal place if the standard deviation is higher than 1? It must be 51 ± 2.

A13. Decimal place of the average values was the same to each datum. The error range (±) along with the average value is not the standard deviation.

Q14. Figure 3: the writings at the base of the images are illegible, so I suggest either to remove them (and put the information in the figure caption) or to improve the resolution and enlarge them.

A14. The writings at the base of the images are the analytical information along with the image served from the instrument. They could be ignored.

Q15. Figure 6: the figures in this flow diagram add not much more, considering are scarcely visible.

A15. All steps were required for explanation of preparation and characterization of the model TRWPs. The mixing steps and the other preparation steps and characterization methods were distinguished by the different colors.

Q16. Typos marked by the reviewer.

A16. Thanks for the marks of typos, the typos were corrected and the corrections were marked in blue.

Reviewer 2 Report

Dear Authors,

The worked carried on the "Preparation and characterization of model tire-road wear particles" is relatively Novel. I would recommend publication after Minor revisions.

  1. Please further enhance the literature review, for a single sentence citations [1-10] are added, the 10 references cited may contain sufficient data to improve the literature review and will best explain the problem statement.
  2. Also, please add the compostition of tyre or cross section of tyre and mention which part/section is more prone to wear.
  3. What happens to the tyre on cement or tar road, what is the life of a average tyre and options to recyle or reuse, add few sentences.
  4. How much pollution is generated by wear of tyres, please give some stats.
  5. Does the tyre particle size goes below Nano scale.? If yes then also add effect on human health, you can refer: The effect of nano-additives in diesel-biodiesel fuel blends: A comprehensive review on stability, engine performance and emission characteristics
  6. Can you please explain the cost incurred in preparation of the sample? and the process of preparation of sample in using simple Figure or block dig.
  7. How did you identify the Real TRWPs when the road dust is collected? is the any specific morphology of TRWPs? How did you seperate?
  8. Can you please add the conslcusion section in points and write some future recommendations of the study.
  9. Can you also suggest how to evade the generation of tyre dust?

Author Response

Q1. Please further enhance the literature review, for a single sentence citations [1-10] are added, the 10 references cited may contain sufficient data to improve the literature review and will best explain the problem statement.

A1. The sentences “Tire tread compounds are usually made of rubber, filler (carbon black and silica), curatives, and other additives. TWPs are generated by friction of the tire tread and road surface. Abrasion rate of tire tread on cement road surface was larger than that on asphalt one [22]. TWP contents in road dust were less than 5 wt% [23]. TRWP can be separated from the road dust using density separation [4,24].” were added in the Introduction part. And the references of 22-24 were also added.

Q2. Also, please add the compostition of tyre or cross section of tyre and mention which part/section is more prone to wear.

A2. The sentences “Tire tread compounds are usually made of rubber, filler (carbon black and silica), curatives, and other additives. TWPs are generated by friction of the tire tread and road surface.” were added in the Introduction part.

Q3. What happens to the tyre on cement or tar road, what is the life of a average tyre and options to recyle or reuse, add few sentences.

A3. The sentence “Abrasion rate of tire tread on cement road surface was larger than that on asphalt one [22].” was added along with the reference 22 in the Introduction part.

  1. Allen, J. O.; Alexandrova, O. Tire wear emissions for asphalt rubber and portland cement concrete pavement surfaces: Arizona Department of Transportation Contract KR-04-0720-TRN Final Report 2006.

Q4. How much pollution is generated by wear of tyres, please give some stats.

A4. The sentence “TWP contents in road dust were less than 5 wt% [23].” was added along with the reference 23 in the Introduction part.

  1. Chae, E.; Jung, U.; Choi, S.-S. Quantification of tire tread wear particles in microparticles produced on the road using oleamide as a novel marker: Environ. Pollut. 2021, 288, 117811.

Q5. Does the tyre particle size goes below Nano scale.? If yes then also add effect on human health, you can refer: The effect of nano-additives in diesel-biodiesel fuel blends: A comprehensive review on stability, engine performance and emission characteristics

A5. PM sizes (PM10 and PM2.5) of the tyre wear particles have been reported, but the tyre wear particles below Nano scale has not been reported.

Q6. Can you please explain the cost incurred in preparation of the sample? and the process of preparation of sample in using simple Figure or block dig.

A6. Cost for the model TRWP preparation will very vary depending on the kinds and amounts. Exact cost can not be estimated. The sample preparation process was already described in Figure 6, Figure SI1, and Figure SI12.

Q7. How did you identify the Real TRWPs when the road dust is collected? is the any specific morphology of TRWPs? How did you seperate?

A7. The sentence “TRWP can be separated from the road dust using density separation [4,24].” was added along with the reference 24 in the Introduction part.

  1. Kovochich, M.; Parker, J. A.; Oh, S. C.; Lee, J. P.; Wagner, S.; Reemtsma, T.; Unice, K.M. Characterization of individual tire and road wear particles in environmental road dust, tunnel dust, and sediment: Environ. Sci. Technol. Lett. 2021, 8, 1057−1064.

Q8. Can you please add the conslcusion section in points and write some future recommendations of the study.

A8. The sentence “Model TRWPs can be used for tracing study of TRWPs produced on the road to sediment of river and sea and the risk for human health.” was added in the Conclusion part.

Q9. Can you also suggest how to evade the generation of tyre dust?

A9. The sentence “The ways to reduce generation of the TWPs would be development of high abrasion-resistant tread compounds and improvement of road conditions.” was added in the Conclusion part.

Reviewer 3 Report

The manuscript (polymers-1655723) presents the preparation of model tire tread wear to particles similar to those found in the environment. The manuscript is interesting and logically arranged, however before publication I have a couple of remarks/suggestions:

  1. Please indicate which of the TGA results were obtained in oxidative conditions.
  2. SEM images quality should be improved and a clearly visible scale bar should be added to all images (including supporting Info)

Author Response

Q1. Please indicate which of the TGA results were obtained in oxidative conditions.

A1. TGA analysis was performed under inert (N2) and oxidation 96 (air) conditions at 50 - 700oC and 700 - 900oC ranges, respectively. This TGA analysis conditions were described in Lines 96-97.

Q2. SEM images quality should be improved and a clearly visible scale bar should be added to all images (including supporting Info).

A2. Sizes of the scale bars were added in the figure captions of Figures 1, 2, 3, and 5. The scale bars are 60 and 6 mm for the magnifications of 200 and 2,000, respectively.

Round 2

Reviewer 1 Report

The new version is certainly improved, but a concern remains as regards the presentation of the data: 

Q. Tables 2 and 3: what is the meaning of reporting decimal place if the standard deviation is higher than 1? It must be 51 ± 2.

A. Decimal place of the average values was the same to each datum. The error range (±) along with the average value is not the standard deviation.

Q. You can name as you want, but if the error is equal to 3 what is the meaning of reporting a decimal place lower than the error? It means that the value is wrong. Thus, like for instance, 54.3±3.1 must be replaced with 54 ± 3

Author Response

Q. You can name as you want, but if the error is equal to 3 what is the meaning of reporting a decimal place lower than the error? It means that the value is wrong. Thus, like for instance, 54.3±3.1 must be replaced with 54 ± 3.

A. The standard deviations were calculated and applied to the data in Tables 2 and 3.